# Novel Monoclonal Antibodies 1D2 and 4E4 Against *Aspergillus* Glycoprotein Antigens Detect Early Invasive Aspergillosis in Mice

**DOI:** 10.3390/jof10120832

**Published:** 2024-12-02

**Authors:** Xihua Lian, Amy Scott-Thomas, John G. Lewis, Madhav Bhatia, Stephen T. Chambers

**Affiliations:** 1Department of Pathology and Biomedical Science, University of Otago, Christchurch 8140, New Zealand; xihua.lian@fjmu.edu.cn (X.L.); john.lewis@otago.ac.nz (J.G.L.); madhav.bhatia@otago.ac.nz (M.B.); 2Department of Ultrasound Medicine, The Second Clinical Medical School, Fujian Medical University, Quanzhou 362000, China

**Keywords:** monoclonal antibody, 1D2, 4E4, *Aspergillus*, invasive aspergillosis, mice

## Abstract

Due to the high morbidity and mortality rates of invasive aspergillosis (IA) and the importance of early IA detection for successful treatment and subsequent outcome, this study aimed to determine a time course of detectable antigen in a mouse model of IA and correlate it with tissue invasion by using two novel monoclonal antibodies, 1D2 and 4E4, that can be used to detect the *Aspergillus*-derived glycoproteins. Immunocompromised mice were randomly divided into five groups: uninfected control, and inoculation with conidia from *Aspergillus fumigatus, Aspergillus flavus*, *Aspergillus niger*, and *Aspergillus terreus*. Conidia (2 × 10^6^ cells/mL) were administered intravenously via tail vein injection. Three mice from each group were euthanised at each time point (6 h, 12 h, 18 h, 24 h, and 48 h) after inoculation. Urine and blood were collected for analysis using a double-sandwich ELISA using 1D2 and 4E4. Liver, spleen, and kidney tissues were harvested for tissue staining. The levels of liver injury in the IA mice progressively increased with time after inoculation with *Aspergillus* conidia. Following inoculation with *A. fumigatus*, swollen conidia were identified in the spleen, as well as antigens in blood after 18 h. Hyphae were detected in the spleen, liver, and kidney after 48 h. For *A. flavus,* the antibodies detected hyphae in the liver and spleen as well as circulating antigens in blood samples 48 h after inoculation. Tissue injury was observed in the mice inoculated with *A. terreus* and *A*. *niger*, but there was no evidence of fungal invasion or antigens in the blood. Antigens were not detectable in mouse urine but could be detected in glomeruli of the kidney by immunofluorescence. In conclusion, the mAb-based antigen detection double-sandwich ELISA results were consistent with the IHC results in this study. Novel monoclonal antibodies 1D2 and 4E4 can serve as tools for the early identification of IA in mice infected by *A. fumigatus* and *A. flavus*. This study also suggests the potential usefulness of this approach in human disease.

## 1. Introduction

Invasive aspergillosis (IA) is one of the most severe disorders of invasive fungal disease [1]; it is most commonly (approximately 90% of cases) caused by *Aspergillus fumigatus*, followed by *Aspergillus flavus*, *Aspergillus niger*, and *Aspergillus terreus* [2]. The morbidity and mortality rate of IA remains high in those who receive immunosuppressive drugs [3], neutropenic patients [4], and patients suffering from severe influenza or coronavirus (COVID-19) [4,5,6] despite treatment with antifungal agents. To date, early-stage IA diagnosis remains a major challenge owing to sub optimal sensitivity and specificity of current diagnostic methods [2,4,7]. Delayed diagnosis results in delayed therapy, leading to an increase in morbidity and mortality for patients [2,4,7]. Given the specific binding between the monoclonal antibody (mAb) and the target antigen, mAb-based methods have paved a new way for assisting the rapid, early, and non-invasive diagnosis of IA [8,9,10]. Early studies using mAb 1D2 and 4E4 indicated the hyphal antigens were detected from all four *Aspergillus* species in filtrates of culture media after a two-week incubation [11]. 

IA animal models play a critical role in improving the translation of a new diagnostic or therapeutic approach prior to use in the clinical setting [9,12,13,14,15]. A successful animal model is central to understanding the capability of a novel method for diagnosing or treating IA [12,14,16]. We investigated two monoclonal antibodies (mAbs) 1D2 and 4E4 against glycoprotein antigens and showed they can detect circulating antigens in mice 48 h post-infection with *A. fumigatus*. Due to the importance of early IA detection for successful treatment and subsequent outcome, we aimed to determine a time course of detectable antigen in a mouse model of IA and correlate it with tissue invasion.

## 2. Materials and Methods

### 2.1. Fungal Strains and Culture Conditions

*A. fumigatus* (AF 293) and *A. flavus* (NRRL 3357) were obtained from the American Type Culture Collection (ATCC). *A. niger* and *A. terreus* strains were clinical strains provided by Canterbury Health Laboratories, Christchurch, New Zealand. All fungal strains were cultivated on Sabouraud dextrose agar (SDA) plates or in Sabouraud dextrose (SD) liquid media at 37 °C.

### 2.2. Preparation of Aspergillus Conidial Suspension

*Aspergillus* conidial suspensions were prepared as previously described with minor changes [11]. Briefly, all *Aspergillus* species were sub-cultured on SDA plates and washed with phosphate-buffered saline (PBS) containing 0.1% Tween 20 (PBST) to acquire conidia. The obtained conidial suspensions were filtered and washed three times. Finally, the conidial precipitates were resuspended with PBS and concentration adjusted to 2 × 10^6^ conidia/mL.

### 2.3. The Production, Purification and Biotinylation of Monoclonal Antibodies

The generation and biotinylation of mAb 1D2 and 4E4 was described previously [11], and they were purified by a mouse IgM purification resin column (LT-145, LigaTrap Technologies, Raleigh, NC, USA) according to the manufacturer’s guidelines. Antibody concentrations were calculated according to the absorbance at 280 nm measured by a UV–visible spectrophotometer. A control mouse IgM antibody (6B10) was purified by the similar method. 

### 2.4. Preparation and Separation of Aspergillus Hyphal Antigens

The *Aspergillus* hyphal antigens were prepared as reported previously [11]. In brief, cell wall-related proteins (CWPs) of *Aspergillus* were precipitated from the cell wall fragments (CWFs) by addition of cold ethanol. Molecular weight cut-off (MWCO) ultrafiltration tubes (3000 MWCO, 10,000 MWCO, 30,000 MWCO, 50,000 MWCO, and 100,000 MWCO) (GE Healthcare, Waukesha, WI, USA) were utilised to separate the CWFs and CWPs into different molecular weights. All filter tubes were pre-treated with sterile PBS to wash and wet the membrane. The samples of CWFs or CWPs were added to different MWCO sizes tubes and centrifuged at 14,000× *g* for 10 to 30 min depending on the MWCO of the membrane. The ultrafiltrate samples were collected in a microcentrifuge tube and stored at −20 °C for future usage. 

### 2.5. Double-Sandwich Enzyme-Linked Immunosorbent Assay (ELISA)

The double-sandwich ELISA was performed similarly to that previously described with minor changes [11]. Briefly, mAb 1D2 (5 µg/mL) was used as the capture antibody. After washing and blocking, 100 μL of each sample or ultrafiltrate samples containing different sizes of CWPs was added and incubated for one hour. Afterward, 4E4-biotin (5 µg/mL) was added, followed by addition of streptavidin-HRP (Jackson ImmunoResearch, Philadelphia, PA, USA) at 1 µg/mL. The signals were detected by addition of tetramethybenzidine (TMB) substrate.

### 2.6. Immunohistochemical Staining and Immunofluorescent Staining

Immunohistochemical (IHC) staining of infected tissues from IA mice was performed as described previously [11]. Tissue fixation, embedding, deparaffinisation and rehydration, antigen retrieval, permeabilisation, blocking, and primary antibody incubation steps for immunofluorescent (IF) staining were similar with IHC but without inactivation of endogenous peroxidase. After three washes, the IF tissue samples were incubated with goat anti-mouse IgM-FITC (Invitrogen, Auckland, New Zealand) at 7.5 µg/mL for one hour at room temperature in the dark, while the IHC samples were incubated with goat anti-mouse IgM-HRP (Abcam, Cambridge, UK) at 0.5 µg/mL. Blocking buffer or isotype IgM 6B10 [17] were used in control groups, but samples were otherwise treated the same. The IF slides were mounted with ProLong™ Glass Antifade Mountant (Invitrogen, Auckland, New Zealand) and observed via an epifluorescence microscope (Carl Zeiss, Oberkochen, Germany). The IHC slides were mounted with Permount™ Mounting Medium (Thermo Fisher Scientific, Auckland, New Zealand) and observed using a light microscope (Olympus corporation, Tokyo, Japan).

### 2.7. The Mouse Model of Invasive Aspergillosis

Animal experiments were carried out in compliance with the New Zealand animal welfare regulation and approved by the Animal Ethics Committee of the University of Otago (AUP-21-145). Male Balb/c mice aged from 10 to 12 weeks were randomly divided into five groups, immunocompromised, *A. fumigatus* infected, *A. flavus* infected, *A. niger* infected, and *A. terreus* infected. All mice were intraperitoneally administrated with two doses of cyclophosphamide (Baxter Healthcare Limited, Auckland, New Zealand) at 150 mg/kg on day one and day four to induce immunosuppression. To establish the IA model, the immunocompromised mice were injected intravenously with 0.1 mL of *A. fumigatus*, *A. flavus*, *A. niger*, or *A. terreus* conidia suspension at 2 × 10^6^ cells/mL on day five. The control group mice was inoculated with 0.1 mL of sterile saline. In each infection group, three mice were euthanised at varying time points (6 h, 12 h, 18 h, 24 h, and 48 h) after inoculation. Urine and blood (heparinised) were collected alongside liver, spleen, and kidney. Given that the neutropenic mice have high risks of bacterial infection, all mice received enrofloxacin (5 mg/kg) subcutaneously as a prophylaxis daily from the first dose of cyclophosphamide to the day they were euthanised.

### 2.8. Statistical Analysis

Statistical analysis was performed with SPSS software (version 21.0, IBM Analytics, New York, NY, USA) and GraphPad software (version 9, Prism, San Diego, CA, USA). All continuous variable results were expressed as [mean ± standard deviation (SD)]. Comparisons were evaluated by independent samples *t*-test or one-way ANOVA with post hoc Tukey’s test. Differences were considered significant when *p* value < 0.05.

## 3. Results

### 3.1. Tissue Samples from the Mouse Model

Gross examination of organs taken from mice after inoculation with *A. fumigatus* showed changes only in the liver specimens harvested 48 h post conidia inoculation. The liver showed evidence of haemorrhage on some surfaces and pallor in others, suggesting infarction and infection (Figure 1).

### 3.2. Microscopic Examination of Tissues

#### 3.2.1. Haematoxylin and Eosin (HE) Staining

Liver tissues from mice inoculated with *A. fumigatus* and *A. flavus* showed tissue injury that became progressively more severe 6, 12, 18, 24, and 48 h after inoculation when compared with the control group (Figure 2). 

Mice in the control group were immunosuppressed but not inoculated with *Aspergillus* conidia. Liver tissues of immunosuppressed mice showed hepatocytes arranged in a radiation shape surrounding the round or oval central hepatic vein without red blood cells inside. The hepatocytes were normal in shape (cubic) with a clear cell nucleus in the centre. For mice infected by *A. fumigatus* and *A. flavus*, six hours post-conidial inoculation, there was no significant necrosis in the tissues. From 12 to 24 h, the liver progressively showed cytoplasmic vacuolation and necrosis of hepatocytes, indicating a progression of inflammation and injury in the liver caused by infection. After 48 h, the tissue showed significant injuries due to the fungal infection. The hepatic lobule structure was disordered, the radial structure disappeared, and there was confluent necrosis. The central hepatic vein was expanded, congested, and full of red blood cells. The hepatocytes were deformed and partly fused with a shrinking or disappearing nucleus. In addition, the *Aspergillus* hyphae infiltrated the blood vessels, but with no obvious neutrophil infiltration.

In parallel, liver tissues from mice inoculated with *A. niger* and *A. terreus* showed similar injurious features between 6 and 24 h after conidia inoculation as those inoculated with *A. fumigatus-* and *A. flavus*-infected mice. However, examination of the liver samples harvested 48 h post-inoculation showed progressive cytoplasmic vacuolation and necrosis of hepatocytes but no hyphal infiltration in the liver (Figure 2 and Appendix A).

#### 3.2.2. Immunohistochemistry (IHC) Staining

No fungal elements were detectable in the tissues from control mice or mice inoculated with *A. fumigatus* conidia in samples harvested at 6 and 12 h, as well as *A. flavus* between 6 and 24 h, post-inoculation. Both 1D2 and 4E4 recognised the cell wall of hyphae of *A. fumigatus* and *A. flavus* in formalin-fixed paraffin tissue sections from IA mice (Figure 3 and Figure 4), indicating that both monoclonal antibodies could bind to the cell wall antigens of *A. fumigatus* and *A. flavus* after they swelled or germinated in mouse tissue. Both antibodies identified the swollen *A. fumigatus* conidia in spleen after 18 h inoculation. Additionally, they recognised the hyphal wall of *A. fumigatus* and *A. flavus* in the liver, spleen, and kidney 48 h post-inoculation. However, there was no positive staining in the tissues from the IA mice infected with *A. niger* and *A. terreus* or the controls at any time post-inoculation. Also, no staining was observed in negative controls with blocking buffer or isotype IgM 6B10.

#### 3.2.3. Kidney Immunofluorescence

Immunofluorescence (IF) of kidney tissues from *A. fumigatus*-infected mice showed positive fluorescence staining in the glomerulus with both 1D2- and 4E4-tagged antigens (Figure 5). This suggested that some antigens were unable to penetrate the glomerular filtration barrier and subsequently remained trapped in the glomerulus.

### 3.3. Detection of Antigens in the Blood and Urine of Mice with Invasive Aspergillosis 

The detection of hyphal cell wall antigens in serial dilutions of blood samples are shown in Figure 6. Antigens were detected in plasma samples from mice 18, 24, and 48 h post-inoculation with *A. fumigatus* conidia and 48 h post-inoculation with *A. flavus* conidia. Samples taken at times prior to this, including baseline, were negative. Interestingly, the titre of the 48 h plasma sample for *A. flavus* blood was the highest observed, despite a negative sample at 24 h. No *Aspergillus* antigen was detected in the plasma from those mice infected with *A. niger* and *A. terreus*. All urine samples were negative in both the inoculated and control mice. 

### 3.4. Correlation of Histologic and ELISA Findings

The results of histological findings and the ELISA assay are summarised in Table 1. These results showed that the antigens became detectable in blood in parallel with detectable *A. fumigatus* swollen spores in splenic tissue at 18 h. However, the blood antigens preceded detectable hyphae in liver and kidney tissues. In contrast, antigens became detectable in blood at 48 h after *A. flavus* inoculation, which followed the same timeline as the detection of hyphae in all the tissues. 

### 3.5. The Molecular Weight of Antigens Recognised by mAb 1D2 and 4E4

Cell wall proteins (CWPs) and cell wall fragments (CWFs) from various MWCO centrifugal filters showed differing reaction signals (Figure 7). Figure 7 shows that there was little reactivity in fractions with a molecular weight (MW) less than 30 kDa in CWP samples that were identified by 1D2 (Figure 7a) and 4E4 (Figure 7b). This demonstrated that most of the antigens had a MW more than 30 kDa. Of note, the antigen concentrations of both CWP and CWF samples gradually increased when they were collected from the 50 kDa and 100 kDa filter tubes. Together, these results indicated that the MW of antigens recognised by 1D2 and 4E4 varied from 30 kDa to 100 kDa, especially between 50 kDa and 100 kDa. Additionally, the concentration of antigens in CWP samples was significantly higher than those in CWF samples (*p* < 0.001), which meant there were more antigens in the CWP sample and the antigens were more likely to be a protein.

## 4. Discussion

The current study confirms some of our previous results and adds to our knowledge on the potential value of the two monoclonal antibodies for the diagnosis of invasive aspergillosis by determining the time course for antigens to appear in the peripheral blood of immune-suppressed mice, the size of the cell wall proteins, and a possible reason why the antigen was not detectable in urine. Our results demonstrated that there was a progressive increase in severity of liver injury with time post-inoculation with *Aspergillus* conidia in the mice. The earliest evidence of fungal invasion was found in samples taken 18 h after *A. fumigatus* conidia inoculation, when swollen conidia were seen in splenic tissue samples and antigen was detected in the blood. *A. fumigatus* hyphae were present in tissue samples harvested at 48 h after conidia injection. In contrast, fungal elements were not detected in tissue samples until 48 h after *A. flavus* conidia inoculation, when they were identified in both the liver and spleen alongside antigen-positive blood samples. While there was tissue injury in mice inoculated with *A. terreus* and *A. niger*, there was no evidence of hyphal invasion or antigen present in blood. These results demonstrated that the level of antigen in blood correlated with germination of *A. fumigatus* conidia and the presence of *A. flavus* hyphae in the tissues.

The immune-compromised mouse model used within this study allowed for the development of IA using *A. fumigatus* and *A. flavus* conidia, which are the two most common species to cause human disease. Invasive infection in the mice was preceded by tissue injury that may be partially related to the cyclophosphamide-induced neutropenia, although mycotoxins produced by the *Aspergillus* species may have contributed to this. The described mouse model was less useful for *A. niger* and *A. terreus* as significant tissue invasion was not detected, though evidence of inflammation was detected. This may have been because tissues were harvested 48 h after inoculation, and tissue invasion may have occurred at later time points, reflecting the susceptibility of the mouse to the strain used [18]. In addition, 12 h post-inoculation with *Aspergillus*, the liver tissue showed cytoplasmic vacuolation. Forty-eight hours after inoculation with *A. fumigatus* and *A. flavus*, necrosis of hepatocytes was more severe, indicating a progressive inflammation and injury in the liver in mice. However, we did not detect significant abnormity in lung tissues in this study; this might have been because we induced the IA model via mouse tail vein, and 48 h is not enough time for *Aspergillus* to penetrate the blood-air barrier to cause pulmonary infection and injury. 

The 1D2 and 4E4-biotin pair assay was able to detect the antigen in extracellular fractions in culture media from *A. niger* and *A. terreus* as well as *A. fumigatus* and *A. flavus*, although the amount of antigen in the supernatant of *A. niger* and *A. terreus* was lower compared to *A. fumigatus* and *A. flavus* [11]. In a similar way, this pair of mAb-based ELISA detected antigens in the blood from the early stage of IA. Eighteen hours after inoculation with *A. fumigatus*, *Aspergillus* antigens were released into the bloodstream. This suggested that soluble antigens were more rapidly secreted into the blood after inoculation with *A. fumigatus* compared to *A. flavus*. Moreover, the antigen was detected in the blood 18 h after inoculation, earlier than other studies that report detection at least 24 h after inoculation using monoclonal antibodies to galactomannan and (1→3)-β-D-glucan [19,20,21]. On the other hand, in *A. flavus*-infected mouse samples, this double-sandwich ELISA detected *A. flavus* antigens in blood 48 h post-inoculation. Interestingly, this signal was significantly higher compared to *A. fumigatus*, indicating that *A. flavus* starts to secrete abundant soluble antigens into the blood 48 h post-inoculation, as there were no antigens detected before that time. The blood analysis studies of samples from mice inoculated with *A. niger* or *A. terreus* showed no evidence of tissue invasion or antigen in the blood, which was consistent with IHC staining of tissues. Our previous study showed immunofluorescent staining of hyphae were negative despite the antigen being detectable in the supernatant of cultures. This may have been due to lower levels of antigen in the hyphal or masking of the epitope from different orientations of the antigen. In the current study, the negative results may result from the less pathogenic characteristics of both *A. niger* and *A. terreus* when compared to *A. fumigatus* and *A. flavus,* which failed to establish a significant infection in this study [18]. It is also possible that the structure of antigens may vary between different species of *Aspergillus,* and this might lead to the different response time or its negative results in some species. Though this study has demonstrated more usefulness of 1D2 and 4E4 in diagnosing IA in mice infected by different *Aspergillus* species at different time points when compared to the previous study [11], future work needs to be done to observe mice infected with established infection with *A. niger and A. terreus* for a longer period of time and to determine if antigens can then be detected in the blood and tissue samples. Overall, these results suggest that the ELISA assay in the plasma performed well, and they reported a positive result early in the course of infection that was also easy to perform. This relatively non-invasive diagnostic test may have potential for the early diagnosis of IA in human patients who are the high-risk IA group.

We were unable to detect the antigen in urine samples from any of the mice. Immunofluorescence in kidney sections was performed, and positive fluorescence was detected in the glomerulus by 1D2 and 4E4, indicating the antigens might get “stuck” in the kidney and be unable to cross the glomerular wall. The glomerular capillary wall, the essential barrier for kidney filtration, is principally composed of three layers—the capillary endothelial cells, the basement membrane, and the epithelial cells called podocytes [22]. The distinctive pore dimension and negative charge of each layer together account for the selective permeability of the glomerular filtration membrane [22,23]. The molecular weight cut-off value of the glomerular filtration is between 30 kDa and 50 kDa [24,25], and for *Aspergillus* antigens with molecular weight less than 30 kDa, these can cross the glomerular capillary wall so that they can be detected in a urine sample, such as galactofuranose and galactomannan, as has been reported [10,26,27,28]. However, a second mechanism may have contributed to the absence of antigens between 30 kDa and 37 kDa in the urine. Most proteins in the blood are negatively charged and are repelled by the negatively charged glomerular filtration membrane. It is also conceivable that some glycoprotein antigens may penetrate the three-layer filtration barrier but are resorbed in the proximal tubule back into the bloodstream [24,29].

These studies demonstrate that the antibodies readily detect invasive hyphae of *A. fumigatus* and *A. flavus* early in the course of infection and may have potential in the histological diagnosis of human IA with these two important pathogens from biopsy specimens. We were unable to demonstrate hyphae in mice inoculated *A. niger* or *A. terreus* conidia, but these may be detectable later in the course of infection, which we were unable to do. This mAb-based assay is worthy of investigation for application to the diagnosis of IA as the antigen is detectable early in the course of infection with *A. fumigatus*.

## 5. Conclusions

Collectively, 1D2 and 4E4-biotin double-sandwich ELISA detected circulating antigens as early as 18 h after mice were infected with *A. fumigatus* and 48 h after inoculation with *A. flavus*. Moreover, both 1D2 and 4E4 also identified hyphae or swollen conidia in the liver, kidney, and spleen samples from IA mice. Finally, there was no positive antigen in the urine of IA mice that could be detected by mAb 1D2 and 4E4-biotin-based ELISA. These mAbs may have application for detecting circulating antigen in blood, rather than urine, and might have value for the early diagnosis of IA in humans. 

## Figures and Tables

**Figure 1 jof-10-00832-f001:**
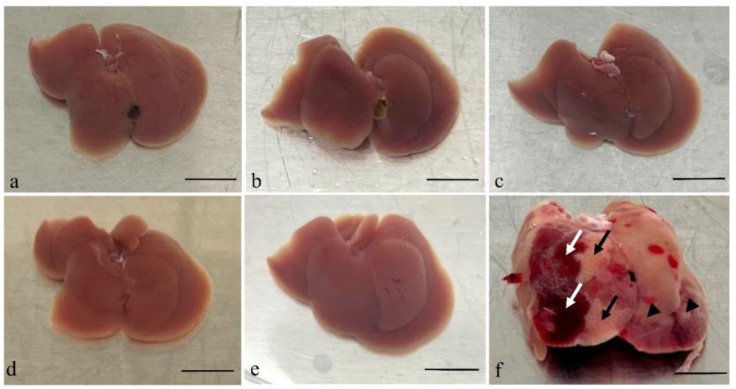
Gross examination of the livers taken after the mice infected by *A. fumigatus* at different time points. Images of liver tissues from an immunocompromised mouse without *Aspergillus* inoculation (**a**) and tissues collected post-inoculation with *Aspergillus* are shown (**b**–**f**). Tissues harvested 6 h (**b**), 12 h (**c**), 18 h (**d**), and 24 h (**e**) post-inoculation showed no significant morphological changes. However, the liver specimen harvested 48 h (**f**) post-inoculation was significantly swollen and showed evidence of infection (black arrow head) and haemorrhage (white arrow) on the surface, as well as pallor suggestive of infarction (black arrow). Scale bars represent 1 cm.

**Figure 2 jof-10-00832-f002:**
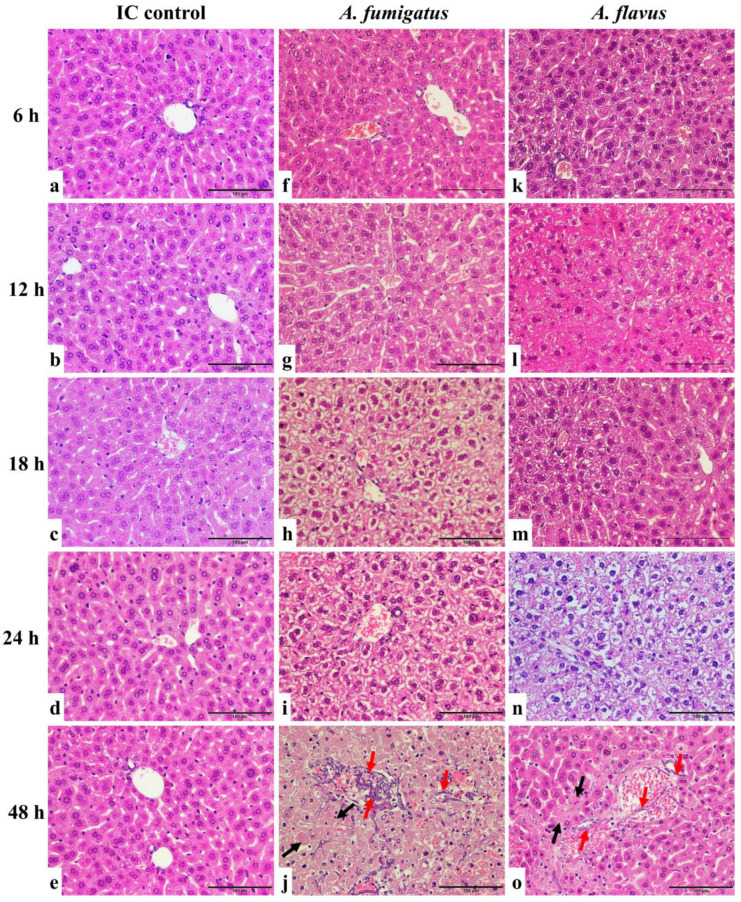
HE staining of liver sections of *A. fumigatus-* and *A. flavus*-infected mice. The liver of immunocompromised (IC) control mice did not show significant tissue injury (**a**–**e**). The samples from *A. fumigatus-*inoculated (**f**–**j**)) and *A. flavus*-inoculated (**k**–**o**) mice showed no significant necrosis six hours post-inoculation (**f**,**k**). From 12 to 24 h, the liver progressively showed cytoplasmic vacuolation and necrosis of hepatocytes (**g**–**i**,**l**–**n**). After 48 h, the tissue showed significant injuries with confluent necrosis (black arrow) and *Aspergillus* hyphae infiltration (red arrow) (**j**,**o**). Scale bars represent 100 μm.

**Figure 3 jof-10-00832-f003:**
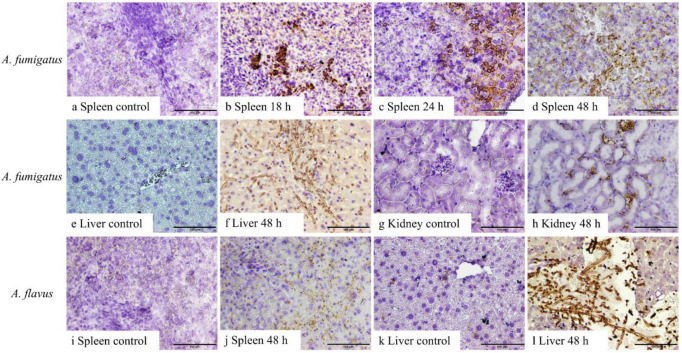
IHC staining of 1D2 in tissue sections of liver, spleen, and kidney from *A. fumigatus-* and *A. flavus*-infected mice. *A. fumigatus*-infected tissues (**a**–**h**), compared to the negative IHC staining in spleen taken 6–12 h after inoculation (**a**), swollen conidial wall (**b**,**c**), and hyphal wall (**d**) (brown staining) seen after 18 h (**b**), 24 h (**c**), and 48 h (**d**) post-inoculation. In addition, compared to the tissues collected 6, 12, 18, and 24 h post-inoculation (**e**,**g**), 1D2 also identified hyphal wall in the liver (**f**) and kidneys (**h**) 48 h post-inoculation. Likewise, for *A. flavus*-infected animals (**i**–**l**), hyphal wall elements were detected in the spleen (**j**) and liver (**l**) 48 h post-inoculation when compared to samples collected at 6, 12, 18, and 24 h post-inoculation (**i**,**k**). Scale bars represent 100 μm.

**Figure 4 jof-10-00832-f004:**
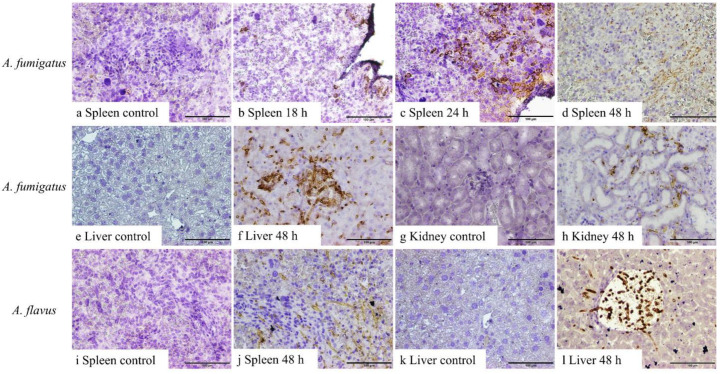
IHC staining of 4E4 in tissue sections of the liver, spleen, and kidney from *A. fumigatus-* and *A. flavus*-infected mice. *A. fumigatus*-infected tissues (**a**–**h**), compared to the negative IHC staining in spleen taken 6–12 h after inoculation (**a**), swollen conidial wall (**b**,**c**), and hyphal wall (**d**) (brown staining) detected after 18 h (**b**), 24 h (**c**), and 48 h (**d**) inoculation. In addition, compared to the tissues collected 6, 12, 18, and 24 h post-inoculation (**e**,**g**), hyphal walls were identified in the liver (**f**) and kidneys (**h**) 48 h post-inoculation. Likewise, for *A. flavus*-infected samples (**i**–**l**), 4E4 can detect the hyphal wall in the spleen (**j**) and liver (**l**) 48 h post-inoculation when compared to samples collected at 6, 12, 18, and 24 h post-inoculation (**i**,**k**). Scale bars represent 100 μm.

**Figure 5 jof-10-00832-f005:**
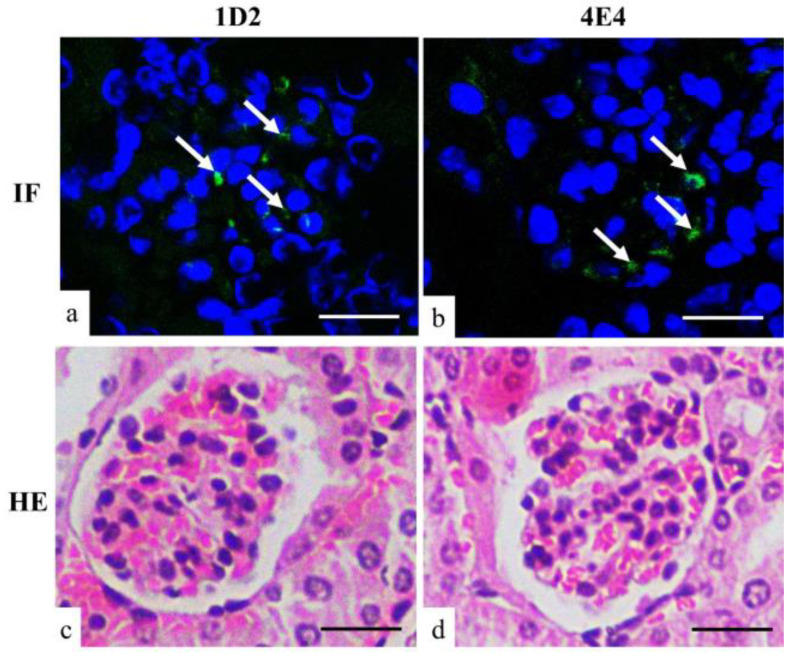
Immunofluorescent (**a**,**b**) and HE (**c**,**d**) staining of mouse kidney. Both 1D2 (**a**) and 4E4 (**b**) showed positive fluorescent signals (arrow) in the glomerulus. Scale bars represent 20 μm.

**Figure 6 jof-10-00832-f006:**
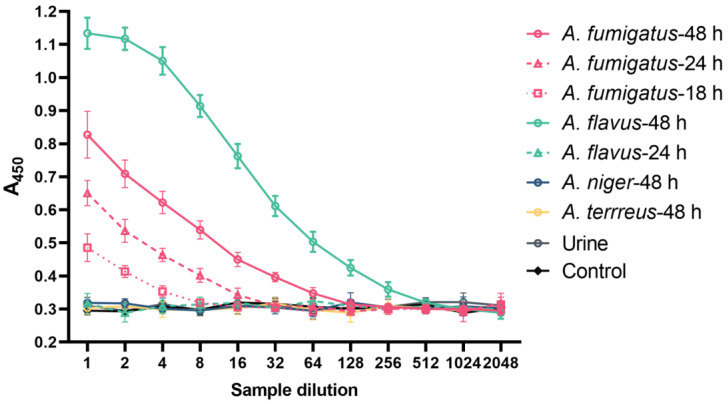
The detection of antigens in the blood and urine of IA mice. The double-sandwich ELISA detects circulating antigens in dilutions of plasma from IA mice (*n* = 3 per time point) infected by *A. fumigatus* (after 18 h, 24 h and 48 h of inoculation) and *A. flavus* (after 48 h of inoculation). The assay was tested in triplicate. Data are presented as mean ± SD. A_450_: absorbance at 450 nm.

**Figure 7 jof-10-00832-f007:**
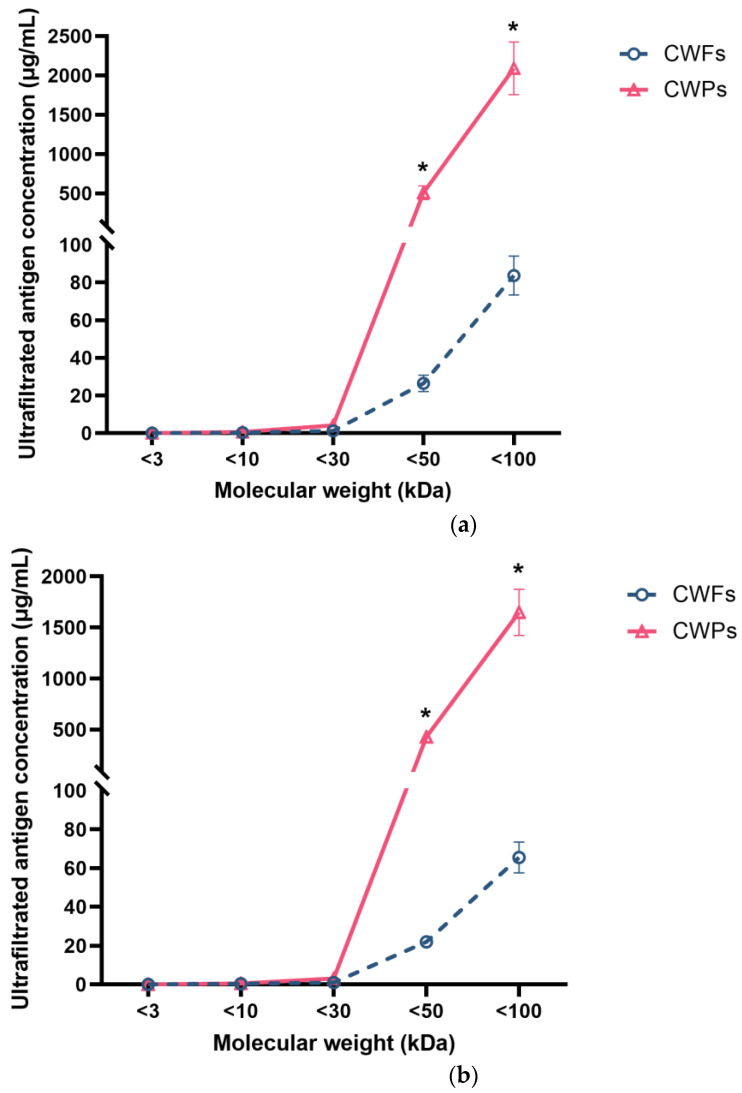
The concentration of ultra-filtered antigens with various molecular weights detected by ELISA using monoclonal antibodies. MAb 1D2 (5 µg/mL) was coated overnight, followed by blocking with 5% BSA; 100 μL of ultrafiltrate samples of cell wall fragments (CWFs) or cell wall proteins (CWPs) were added to the wells. After this, the plate was incubated with 4E4-biotin (5 µg/mL). Little signal was detected in ultra-filtrate samples less than 30 kDa for both 1D2 (**a**) and 4E4 (**b**). The concentration of samples collected from the 50 kDa and 100 kDa filter tubes gradually increased. This assay was undertaken in triplicate. Data were presented as mean ± SD and analysed using a two-tailed Student’s *t* test. *: *p* < 0.001 verse CWPs.

**Table 1 jof-10-00832-t001:** Summary of plasma ELISA, tissue IHC, and HE staining results.

Species	Time (hour)	ELISA	IHC			HE		
			Liver	Spleen	Kidney	Liver	Spleen	Kidney
*A. fumigatus*	6	−	−	−	−	−	−	−
	12	−	−	−	−	−	−	−
	18	+	−	+ *	−	−	−	−
	24	+	−	+ *	−	−	−	−
	48	++	+	+	+	+	+	+
*A. flavus*	6–24	−	−	−	−	−	−	−
	48	+++	+	+	−	+	−	−
*A. niger*	6–48	-	−	−	−	−	−	−
*A. terreus*	6–48	−	−	−	−	−	−	−

*: only swollen spores were stained positive in the tissue. +++: strong positive signal; ++: moderate positive signal; +: positive signal; −: negative signal.

## Data Availability

All data generated or analysed during this study are included in this published article.

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
