# Peer review of "Novel Monoclonal Antibodies 1D2 and 4E4 Against Aspergillus Glycoprotein Antigens Detect Early Invasive Aspergillosis in Mice"

_jof, 2024, doi:10.3390/jof10120832_

Round 1
Reviewer 1 Report
No major comments.
There are no major comments. However, the following points should be considered:
1- Use arrows in Figure 2 to point to the hyphae infiltration and necrosis
2- Figure 3 is repeated. Maybe it is a formatting issue (please make sure it is not duplicating
3- In the discussion, the authors should clarify why they did not use lung specimens in their study.
4- In the discussion, compare the results of these two mAbs with other mAbs used in other studies.
5- What are the major differences between this study and the previous study "Two Monoclonal Antibodies That Specifically Recognize Aspergillus Cell Wall Antigens and Can Detect Circulating Antigens in Infected Mice.". explain that in the discussion. )
Author Response
Reviewer one
We really appreciate for your comments and suggestion on the study. Below is the point-to-point response to your comments.
There are no major comments. However, the following points should be considered:
1- Use arrows in Figure 2 to point to the hyphae infiltration and necrosis
Response: We have added the arrows in Figure 2 accordingly.
2- Figure 3 is repeated. Maybe it is a formatting issue (please make sure it is not duplicating
Response: Figure 3 and Figure 4 demonstrate the IHC staining results of antibody 1D2 and 4E4 respectively, as such, they are not repeated.
3- In the discussion, the authors should clarify why they did not use lung specimens in their study.
Response: We have clarified this in the discussion. See line 303
4- In the discussion, compare the results of these two mAbs with other mAbs used in other studies.
Response: We have discussed the difference in the third paragraph: the antigen was detected by 1D2 and 4E4-biotin ELISA in the blood 18 hours after inoculation, earlier than other studies that report a positive reaction after 24 hours. We have added a comment and references to monoclonal antibodies to galactomannan and (1→3)-β-D-glucan in experimental animal studies. Also, we discussed other differences in another published paper (DOI: 10.3390/ijms23010252). See lines 314-316
5- What are the major differences between this study and the previous study "Two Monoclonal Antibodies That Specifically Recognize Aspergillus Cell Wall Antigens and Can Detect Circulating Antigens in Infected Mice.". explain that in the discussion.
Response: We have discussed this in the third paragraph of discussion. The main differences between this study and the previous study are listed as follows: The previous study focused on nature of the antigen, the distribution of antigen recognized by 1D2 and 4E4 within the hyphal walls by immunofluorescence, the specificity of the antibodies, as well as the potential diagnostic value in IA mice infected by A. fumigatus after 48 hours. On the other hand, this study investigated the time course and potential early diagnostic value of these two antibodies in IA mice infected by different Aspergillus species. The study tests included blood testing, HE staining and IHC staining, determining the size of cell wall proteins a likely explanation as to why antigen was not present in urine samples.
We have added to the discussion - The current study confirms some of our previous results adds our knowledge on the potential value of the two monoclonal antibodies for the diagnosis of invasive aspergillosis by determining the time course for antigen to appear in the peripheral blood of immune suppressed mice, the size of the cell wall proteins and a possible reason why the antigen was not detectable in urine. See lines 275-279
Reviewer 2 Report
Aspergillosis is a common disease that primarily affects people who receive immunosuppressive drugs, neutropenic patients, as well as patients suffering from severe influenza or corona virus. The article "Novel monoclonal antibodies 1D2 and 4E4 against Aspergillus glycoprotein antigens detect early invasive aspergillosis in mice" is very relevant, as it demonstrates the possibility of early detection of Aspergillus antigens using two monoclonal antibodies in a mouse model.
Aspergillosis is a common disease that primarily affects people who receive immunosuppressive drugs, neutropenic patients, as well as patients suffering from severe influenza or corona virus. The article "Novel monoclonal antibodies 1D2 and 4E4 against Aspergillus glycoprotein antigens detect early invasive aspergillosis in mice" is very relevant, as it demonstrates the possibility of early detection of Aspergillus antigens using two monoclonal antibodies in a mouse model.
While reading the article, several questions and comments arose:
What is the difference in the specificity of the monoclonal antibodies used (different epitopes, presence/absence of a carbohydrate component of antigens, structure of the carbohydrate component of antigens?
What is known about the structure of antigens of different types of Aspergillus: are there differences in glycosylation, in the amino acids of epitopes? Could this be the reason for the different response time or its absence on the monoclonal antibodies?
To what extent can the data on the appearance or absence of antigens in the blood obtained when administered intravenously to mice conidia suspension of A. fumigatus, A. flavus, A. niger, or A. terreus be transferred to the possibility of diagnosing aspergillosis in people in whom aspergillus infection occurs, obviously, in another way?
For A. fumigatus, a gradual increase in the amount of antigens in blood plasma is shown. In the case of A. flavus, there is a sharp increase in the concentration of antigens after 48 hours. What is the reason for such differences?
What is included in the CWF? Does the composition of CWP and CWF proteins differ in samples? Are the proteins included in these samples glycoproteins?
Lines 303-305: the authors note that "the amount of antigen in the supernatant of A. niger and A. terreus was lower compared to A. fumigatus and A. flavus". And what can be said about the content of antigens on the cell surface?

Author Response
Reviewer 2
We really appreciate for your comments and suggestion on the study. Below is the point-to-point response to your comments.
Aspergillosis is a common disease that primarily affects people who receive immunosuppressive drugs, neutropenic patients, as well as patients suffering from severe influenza or corona virus. The article "Novel monoclonal antibodies 1D2 and 4E4 against Aspergillus glycoprotein antigens detect early invasive aspergillosis in mice" is very relevant, as it demonstrates the possibility of early detection of Aspergillus antigens using two monoclonal antibodies in a mouse model.
Detail comments
Aspergillosis is a common disease that primarily affects people who receive immunosuppressive drugs, neutropenic patients, as well as patients suffering from severe influenza or corona virus. The article "Novel monoclonal antibodies 1D2 and 4E4 against Aspergillus glycoprotein antigens detect early invasive aspergillosis in mice" is very relevant, as it demonstrates the possibility of early detection of Aspergillus antigens using two monoclonal antibodies in a mouse model.
While reading the article, several questions and comments arose:
What is the difference in the specificity of the monoclonal antibodies used (different epitopes, presence/absence of a carbohydrate component of antigens, structure of the carbohydrate component of antigens?
Response: The two antibodies recognize different epitopes. We have demonstrated these differences in detail in our previous study (DOI: 10.3390/ijms23010252) but we have not yet established further details on the precise structure of the antigen.
What is known about the structure of antigens of different types of Aspergillus: are there differences in glycosylation, in the amino acids of epitopes? Could this be the reason for the different response time or its absence on the monoclonal antibodies?
Response: At present we do not know whether there are any differences in the antigen produced by different species. This is under investigation. We agree that the structure of antigens from different types of Aspergillus species, including their glycosylation patterns and amino acid sequences, shows notable diversity. These structural variations influence antigenic epitopes, which are critical for immune recognition and specificity and might be the reason for the different response time or negative results between species. We have discussed this in the third paragraph of discussion (See lines 325-330).
To what extent can the data on the appearance or absence of antigens in the blood obtained when administered intravenously to mice conidia suspension of A. fumigatus, A. flavus, A. niger, or A. terreus be transferred to the possibility of diagnosing aspergillosis in people in whom aspergillus infection occurs, obviously, in another way?
Response: This is a good question. No matter in what method is used to induce IA (e.g. in this study we induced mice IA via tail vein and IA in human mostly caused by inhalation through the nose), the final outcome is that the Aspergillus penetrates to blood and causes infection in various organs. Therefore, we believe that our results have important value in translating this knowledge to the diagnosis of aspergillosis in humans. Unfortunately, our animal facility would not allow us to use an inhalation method to induce infection so we could not compare result from these methods.
For A. fumigatus, a gradual increase in the amount of antigens in blood plasma is shown. In the case of A. flavus, there is a sharp increase in the concentration of antigens after 48 hours. What is the reason for such differences?
Response: We have discussed this in the third paragraph of discussion: Eighteen hours after inoculation with A. fumigatus, Aspergillus antigens were released into the blood stream. This suggested that soluble antigens were more rapidly secreted into the blood after inoculation with A. fumigatus compared to A. flavus. Moreover, the antigen was detected in the blood 18 hours after inoculation, earlier than other studies that report detection at least 24 hours after inoculation using monoclonal antibodies to galactomannan and (1→3)-β-D-glucan [19-21]. On the other hand, in A. flavus infected mouse samples, this double sandwich ELISA detected A. flavus antigens in blood 48 hours post inoculation. Interestingly, this signal was significantly higher compared to A. fumigatus, indicating the A. flavus starts to secrete abundant soluble antigens into the blood 48 hours post inoculation, as there were no antigens detected before that time. The reasons for this are not clear but may relate to a different rate of growth between the two species or kinetics of antigen release. Further work will be need to determine the reasons.
What is included in the CWF? Does the composition of CWP and CWF proteins differ in samples? Are the proteins included in these samples glycoproteins?
Response: The Aspergillus hyphae were disrupted by sonication at 4℃ and to collect CWF in the supernatant, so we assume most of the cell wall components were included in the CWF. As CWP was precipitated from the CWF sample to collect the protein component, as such, the composition of CWP and CWF proteins should be same or similar. These samples include different types of proteins including glycoproteins.
Lines 303-305: the authors note that "the amount of antigen in the supernatant of A. niger and A. terreus was lower compared to A. fumigatus and A. flavus". And what can be said about the content of antigens on the cell surface?
Response: Thank you. In our previous studies immunofluorescence staining the CWP hyphae of A fumigatus and A. flavus were strongly positive whereas A. terreus and A. niger were negative. (DOI: 10.3390/ijms23010252). Antigen was detectable in the supernatant of all four species but at lower levels for A. terreus and A. niger. A possible explanation is that the antigen while present is masked by other structures or has a different orientation but the antigen content in the cell wall may also be lower. We have added a brief comment to this effect (see lines 317- 325).